# Analysis of Expression and Regulation of AKR1C2 in HPV-Positive and -Negative Oropharyngeal Squamous Cell Carcinoma

**DOI:** 10.3390/cancers16172976

**Published:** 2024-08-27

**Authors:** Maria Ziogas, Oliver Siefer, Nora Wuerdemann, Harini Balaji, Elena Gross, Uta Drebber, Jens Peter Klussmann, Christian U. Huebbers

**Affiliations:** 1Department of Otorhinolaryngology, Head and Neck Surgery, University Hospital of Cologne, 50937 Cologne, Germany; oliver.siefer@uni-koeln.de (O.S.); harini.balaji@uk-koeln.de (H.B.); jens.klussmann@uk-koeln.de (J.P.K.); christian.huebbers@uk-koeln.de (C.U.H.); 2Molecular Head and Neck Oncology, Translational Research in Infectious Diseases and Oncology (TRIO) Research Building, University Hospital of Cologne, 50937 Cologne, Germany; 3Department of Internal Medicine, Faculty of Medicine, Center for Integrated Oncology Aachen Bonn Cologne Duesseldorf, University Hospital Cologne, 50937 Cologne, Germany; nora.wuerdemann@uk-koeln.de; 4Department of Neurology, University Hospital of Cologne, 50924 Cologne, Germany; elena.gross@uk-koeln.de; 5Institute for Pathology, University Hospital of Cologne, 50937 Cologne, Germany; uta.drebber@uk-koeln.de

**Keywords:** human papillomavirus, oropharyngeal squamous cell carcinoma, aldo-keto-reductase 1 C2, oxidative stress

## Abstract

**Simple Summary:**

Oropharyngeal Squamous Cell Carcinoma (OPSCC) represents a significant fraction of head and neck cancers, with a challenging five-year survival rate of only 50%. Key risk factors include tobacco and alcohol consumption and infection with human papillomavirus (HPV), particularly HPV16. Distinct biological differences exist between HPV-positive and HPV-negative OPSCC, including differences in mutation patterns and gene expression profiles. This study focuses on aldo-keto reductases (AKRs), specifically AKR1C2, which are involved in cellular stress management and detoxification processes, particularly in cisplatin-resistant tumors. This study investigates the role of AKR1C2 in HPV-positive OPSCC and its effect on patient outcomes. The findings indicate that increased levels of AKR1C2 are linked to unfavorable prognosis, particularly in male patients, while higher levels in female patients indicate a favorable prognosis.

**Abstract:**

Head and Neck Squamous Cell Carcinoma (HNSCC), particularly Oropharyngeal Squamous Cell Carcinoma (OPSCC), is a major global health challenge due to its increasing incidence and high mortality rate. This study investigates the role of aldo-keto reductase 1C2 (AKR1C2) in OPSCC, focusing on its expression, correlation with Human Papillomavirus (HPV) status, oxidative stress status, and clinical outcomes, with an emphasis on sex-specific differences. We analyzed AKR1C2 expression using immunohistochemistry in formalin-fixed, paraffin-embedded tissue samples from 51 OPSCC patients. Additionally, we performed RT-qPCR in cultured HPV16-E6*I and HPV16-E6 overexpressing HEK293 cell lines (p53^WT^). Statistical analyses were performed to assess the correlation between AKR1C2 expression and patient data. Our results indicate a significant association between increased AKR1C2 expression and higher AJCC classification (*p* = 0.009) as well as positive HPV status (*p* = 0.008). Prognostic implications of AKR1C2 varied by sex, whereby female patients with high AKR1C2 expression had better overall survival, whereas male patients exhibited poorer outcomes. Additionally, AKR1C2 expression was linked to HPV status, suggesting a potential HPV-specific regulatory mechanism. These findings underscore the complex interplay among AKR1C2, HPV, and patient sex, highlighting the need for personalized treatment strategies for OPSCC. Targeted inhibition of AKR1C2, considering sex-specific differences, may enhance therapeutic outcomes. Future research should investigate these mechanisms to enhance treatment efficacy.

## 1. Introduction

Head and Neck Squamous Cell Carcinoma (HNSCC) is the sixth most common malignancy worldwide, with an estimated 878,000 new cases reported in 2022 [1]. Squamous Cell Carcinoma of Oropharyngeal Origin (OPSCC) is a subgroup of HNSCC. Although treatment has improved in recent years, the global mortality rate remains high. Approximately 50% of patients with OPSCC survive the first five years after diagnosis [2,3].

The most common risk factors for the development of HNSCC, in general, are tobacco smoke, often in combination with excessive alcohol abuse [4], and infections with high-risk human papillomavirus (HPV) genotypes, with HPV16 being particularly associated with OPSCC [5]. The Global Cancer Observatory (GLOBOCAN) predicted an even greater increase in incidence, with an estimated rate of 30%. This effectively means that we would be faced with 1.08 million new cases per year by 2030 [6,7]. Data from the United States show that the incidence of HPV-associated OPSCC already surpassed that of HPV-positive cervical cancer [8,9,10].

Since HPV-positive and HPV-negative OPSCC show different clinicopathological characteristics, as well as biological profiles, mutation patterns, and expression signatures, the TNM Classification of OPSSC has been adapted accordingly to distinguish between these two groups. Using the surrogate marker p16^INK4a^, HPV-related (p16^INK4a^-positive) and HPV-negative (p16^INK4a^-negative) OPSCC can be differentiated, thus, different prognoses can be considered [11]. However, the benefits of different treatment strategies in patients with HPV-associated and HPV-non-associated OPSCC are still being discussed [12]. A subpopulation of 20–25% of HPV-positive OPSCC patients present with poor prognosis due to locoregional recurrence or metastatic disease, which may be linked to additional risk factors such as smoking, EGFR overexpression, advanced nodal stage, and chromosomal instability [13,14,15,16]. Strikingly, the risk for women developing HPV-positive OPSCC is approximately four times lower than for men [17]. A possible explanation for this apparent discrepancy could be the different hormonal signals between men and women, which have been discussed as cofactors for HPV-related cancers [18].

This study analyzes the role of aldo-keto reductases (AKR), with a particular focus on AKR1C2. AKR1Cs are important in the epithelial response to oxidative stress. Together with other associated proteins, such as its family members AKR1C1, AKR1C3, NADPH oxidoreductase (quinone 1) (NQO1), superoxide dismutase (SOD1), and haem oxygenase (HQ), it belongs to the group of genes controlled by antioxidant response elements (ARE), which are increasingly expressed in the case of electrophilic or oxidative stress. The expression of ARE element-containing genes is coupled to the Nrf2-KEAP1-CUL3 pathway and therefore predominantly dependent on the regulatory function of these upstream proteins [19]. We previously showed that the upregulation of the ARE element-induced genes AKR1C1 and AKR1C3 correlates with poor prognosis in patients with oropharyngeal carcinomas and is associated with the oxidative stress response system [20].

AKR1C1, AKR1C2, and AKR1C3 metabolize lipids, including steroid hormones, and serve as phase I detoxification enzymes, enabling them to metabolize exogenous substrates [19]. For example, AKR1C2 affects carcinogenesis and prognosis by reducing one of the strongest nitrosamine carcinogens in tobacco, nicotine-derived nitrosaminoketone (NNK), into its detoxified substrate, nitrosamine alcohol (NNAL) [21]. Furthermore, the upregulation of AKR1C2 and its family members prevents the accumulation of reactive oxygen species (ROS) and ROS-derived peroxides, such as cytotoxic lipid oxides [22]. Cisplatin is typically used in adjuvant therapy and in the treatment of non-operable OPSCC. It has been proposed that cisplatin induces cytotoxic lipids such as 4-hydroxynonenale (4-HNE), and cellular stress-induced upregulation of AKR1Cs might prevent their intended accumulation in tumor cells [22]. Furthermore, AKR1C2 is a type 3 hydroxysteroid dehydrogenase that transforms steroid hormones such as progesterone, testosterone derivates, and estrogen [23,24]. Interestingly, AKR1Cs generate feedback loops, amplifying their own expression by controlling NRF2 expression. For example, the conversion of estradione (E1) to 17β-estradiol (E2) by AKR1Cs promotes NRF2 expression and results in the induction of estrogen receptors such as the estrogen-related receptor ⍺ (ERR⍺) [25,26]. The involvement of sex hormone levels and metabolism and the resulting sex- and patient-specific differences may be underlined by the common observation that women develop OPSCC less frequently. Moreover, based on the clinical data of 1629 OPSCC patients, we have recently reported that women present with significantly longer overall survival than men [27].

In the present study, we aim to investigate the correlation between HPV infection, AKR1C2 expression, and oxidative stress mechanisms in relation to clinical data, such as sex, TNM classification, and survival in OPSCC patients. The viral splice product E6*I of the HPV16-E6 protein increases AKR1C1 and C3 expression by binding to their promoter regions [28]. Although AKR1C2 has many similarities to AKR1C1 and AKR1C3, its expression is regulated by independent mechanisms. In addition, AKR1C2 has specific enzyme characteristics, and not all its functions are yet known. 

## 2. Materials and Methods

### 2.1. Subjects and Materials

Formalin-fixed, paraffin-embedded (FFPE) tissue samples from 51 Oropharyngeal Squamous Cell Carcinomas of patients treated at the Department of Otorhinolaryngology and Head and Neck Surgery of the University Hospital of Cologne, Germany, between 2004 and 2011 were analyzed. HPV status was determined using routine PCR and p16^INK4a^ immunohistochemical staining. In total. 25 (49%) samples were HPV-negative and 26 (51%) were HPV-positive (Table 1). AKR1C1, AKR1C3, and NRF2 expression levels were determined in previous studies based on the same cohort and were included in this analysis [20]. 

### 2.2. Ethics Statement

Patient material was used according to the code for proper secondary use of human tissue. The ethics committee of the Medical Faculty of the University of Cologne approved this study (approved protocol no. 11-346). Written informed consent was obtained from all patients.

### 2.3. Immunochemistry 

Immunohistochemical staining was performed on 4 μm thick FFPE tissue sections according to routine protocols using indirect immunolabelling with DAB detection. AKR1C2 expression was detected using rabbit polyclonal antibodies (catalogue number PA5-36572, 1:200 in PBS; Thermo Fisher Scientific, Darmstadt, Germany). 

Briefly, sections were deparaffinized with Roti^®^-Histol (Carl Roth, Karlsruhe, Germany) and rehydrated using a descending alcohol series. Subsequently, the sections were incubated overnight at 70 °C in 0.01 M citrate buffer (pH 6.0) for antigen retrieval, followed by incubation with AKR1C2 antibody at 4 °C overnight. After washing and incubation with corresponding biotinylated goat anti-rabbit secondary antibodies (Vector, Burlingame, CA, USA; 1:250), slides were incubated with avidi-biotin-peroxidase complex (ABC; Vectastatin ABC kit, Vector), and the peroxidase activity was developed with 0.05% 3,3’-diaminobezidine tetrahydrochloride (DAB, Vector) in 0.05 M Tris-HCl (pH 7.6). Sections were mounted in Entellan (Merck, Darmstadt, Germany).

Controls included tumor-free human control tissues selected based on the human protein atlas showing no, moderate, and strong expression, respectively (liver, cervix, and tonsils) [29]. Staining without primary antibodies and IgG isotype controls was negative in all tissues. 

Data on the protein expression of AKR1C1, AKR1C3, and NRF2 were previously published for the same cohort and were included for statistical comparison [20].

### 2.4. Cell Culture and Transfection

HEK293 cells (ATCC: CRL-1573) were grown in DMEM high-glucose medium supplemented with 10% FBS (both Thermo Fisher Scientific, Darmstadt, Germany) under standard conditions (humidified incubator at 37 °C, 5% CO_2_). HPV16-E6 and HPV16-E6*I were cloned from cDNA into pEGFP-N1 vector (Takara Clontech, Saint-Germain-en-Laye, France). Cells were transfected with resulting HPV16-E6*I-GFP, HPV16-E6-GFP, and GFP control vector constructs, respectively, using lipofectamine according to the instructions of the manufacturer (Thermo Fisher Scientific, Germany). Stable HEK293 clones were obtained by selection with 1.2 mg/mL G418.

### 2.5. RT-qPCR Expression Analysis

RT-qPCR was performed as described previously [20]. In brief, RNA was extracted from cultured cell lines using the Qiagen RNeasy mini kit according to the manufacturer’s protocol (Qiagen, Hilden, Germany). In total, 500 ng of total RNA was reverse transcribed (iScript cDNA synthesis kit, BioRad Laboratories, Munich, Germany) and qPCR was performed using iTaq SYBR Green Supermix (BioRad). Amplification was performed using previously described primers applying standard protocols [20]. Hypoxanthine Phosphoribosyltransferase (HPRT) was used for the normalization of mRNA levels.

### 2.6. Statistics

Clinicopathological features were analyzed using cross-tabulations, the χ^2^ test, and Fisher’s exact probability test with SPSS Statistics for Mac version 28.0.1.0 (IBM Software, Armonk, NY, USA). Overall survival rates were estimated over a 5-year period using the Kaplan–Meier algorithm for incomplete observations. Overall survival describes the interval between the date of initial diagnosis and the last date on which the vital status was recorded as “alive” (censored) or the date of death (uncensored). Univariate analyses of variables were performed using the log-rank (Mantel-Cox) test. The minimum sample size for subgroup analysis was determined prior to analysis with a power of 90% and a significance level of 0.05 with 8 samples for each group.

Data analysis was performed using GraphPad Prism 6.0 (GraphPad Software, La Jolla, CA, USA). The significance level was set at *p* < 0.05 for all calculations.

## 3. Results

### 3.1. Immunohistochemical Detection of AKR1C2 

Immunohistochemistry was used to determine the expression of AKR1C2 in epithelial tissues and tumors. In non-tumor tissue, lymphocytes were negative and squamous epithelial keratinocytes showed predominantly no or weak nuclear staining, whereas muscle cells and endothelial cells exhibited strong staining. Control staining without primary antibodies and IgG isotype controls was negative in all tissues The cohort comprised 51 patients, of which 49 FFPE samples with sufficient material were available. Consistent with our previously observed expression pattern for AKR1C1 and AKR1C3 [20,30], staining against AKR1C2 was also positive in adjacent non-tumorous squamous epithelia. As such, we decided to evaluate the staining intensity of both the tumor and the adjacent epithelium and relate them to each other in further analysis, as previously described for AKR1C1/C3 (Figure 1) [20].

The resulting protein intensity ratios showed 29 OPSCC (59.2%) with stronger staining in the tumor compared to the adjacent epithelium (AKR1C2^HIGH^), and the remaining 20 OPSCC (40.8%) showed lower staining than the adjacent epithelium (AKR1C2^LOW^) (Table 1). Moreover, 25 OPSCC were HPV negative (49%), of which 5 presented AKR1C2^LOW^ (10.2%), whereas 15 were AKR1C2^HIGH^ (30.6%). In the case of the 26 HPV-positive OPSCC (51%), 15 presented AKR1C2^LOW^ (30.6%) and 10 AKR1C2^HIGH^ (20.4%) (χ^2^ = 0.008). Furthermore, increased AKR1C2 expression correlated with higher AJCC classification (χ^2^ = 0.009). No significant correlation was found for any of the other parameters analyzed, including alcohol and tobacco consumption and sex.

### 3.2. AKR1C2 Protein Expression, HPV, and Survival in OPSCC

AKR1C2 protein expression was correlated with survival outcomes in combination with clinicopathological data such as sex, HPV status, tumor status, smoking history, alcohol consumption, and protein expression levels of AKR1C1, AKR1C3, and NRF2 (Table 1). Whereas AKR1C2 expression in general (Hazard Ratio (HR) 0.4953, 95% Confidence Interval (CI) 0.1899–1.209, *p* = 0.1229) and in combination with T-Status (HR = 2.734, 95% CI 0.960–7.790, *p* = 0.3162), N-Status (HR 1.785, 95% CI 0.920–3.464, *p* = 0.5722), smoking habit (HR 1.488, 95% CI 0.433–5.116, *p* = 0.7219), and drinking habits (HR 1.530, 95% CI 0.942–2.483, *p* = 0.1988) did not correlate with significant outcomes, AKR1C2 expression presented with a trend for significant correlation with HPV status (HR 1.818, 95% CI 0.669–4.938 *p* = 0.2129; log-rank trend test *p* = 0.0368). However, when considering patient sex, OS was significantly different (HR 1.235, 95% CI 0.704–2.167, *p* = 0.0151). Remarkably, female sex combined with AKR1C2 positivity was predictive of a more favorable outcome, while low AKR1C2-expressing tumors in males were correlated with a better outcome. Therefore, we performed subgroup analyses for both sexes, considering both AKR1C2 expression levels and HPV status into account. Whereas, in women, AKR1C2^HIGH^ tumors presented with a tendency for beneficial survival regardless of HPV status (HR = 0.333, 95% CI 0.028–3.977, *p* = 0.0350), HPV+/AKR1C2^LOW^ tumors presented with far better survival probability, followed by intermediate outcomes for HPV+/AKR1C2^HIGH^ and HPV−/AKR1C2^LOW^ tumors (HR 2.300, 95% CI 0.734–7.201, *p* = 0.0168). HPV−/AKR1C2^HIGH^ tumors presented with the most unfavorable outcome.

For death within 5 years, as well as higher tumor size, a significant correlation was observed. Of note, HPV status appeared to have a strong correlation with AKR1C2 protein expression (*p* = 0.022) (Figure 2). 

In general, the AKR1C2^HIGH^ group of patients was affected by death earlier than the AKR1C2^LOW^ group. Here, low AKR1C2 tumor staining was correlated with an overall survival of 70–80% by the end of five years. In addition, women in this cohort showed better overall survival when AKR1C2 expression in tumor tissue was higher than that in men with high AKR1C2 expression.

### 3.3. Correlation of AKR1C2 with AKR1C1 and AKR1C3 and NRF2

Considering the expression levels of AKR1C1, AKR1C3, and NRF2, AKR1C2 was found to be an independent predictive factor (AKR1C1 and AKR1C3 HR 1.710, 95% CI 1.084–2.698, *p* = 0.0575; NRF2 HR 0.413, 95% CI, 0.166–1.026, *p* = 0.1096) (Table 1).

### 3.4. Effects of HPV16-E6*I on AKR1C2 mRNA Expression

Stable HPV16-E6- or HPV16-E6*I-overexpressing HEK293 cells (p53^WT^) were analyzed by RT-qPCR for the expression of AKR1C2 and its counterparts AKR1C1 and AKR1C3, respectively. While E6*I- but not E6-overexpressing HEK293 cells showed increased AKR1C1 and AKR1C3 expression, AKR1C2 expression was not affected by E6*I or E6, respectively (Figure 3).

## 4. Discussion 

The human aldo-keto reductase family has recently emerged as a promising marker in various cancers and is a key factor in the development of resistance to radio- and chemotherapy [22]. Resistance mechanisms are based either on direct metabolic involvement or contribution to the elimination of cellular stress (e.g., from reactive oxygen species and lipid peroxides). Of particular interest is the possibility of pharmacologically inhibiting AKR1Cs with easily administered and well-known substances, such as NSAID derivatives, thus preventing therapeutically unfavorable protective mechanisms against cellular stress.

In a previous study, we observed the upregulation of AKR1C1 and AKR1C3 expression in a subgroup of HPV16-positive OPSCC along with upregulated HPV16-E6*I mRNA expression. AKR1C1/C3 overexpression has also been associated with poor prognosis in both HPV-positive and HPV-negative OPSCC subgroups [20,28]. 

AKR1C2 is located in the same genomic region on chromosome 10p15-14 and contains all four aldo-keto reductase family 1 member C genes. The proteins show high sequence homology, namely AKR1C1/AKR1C2 with 98% homology, differing in only 7 amino acids, whereas AKR1C2/AKR1C3 show 87% homology and differ in 43 amino acids [31]. However, AKR1Cs have independent substrate specificities, implying an independent (patho-)physiological role. Furthermore, there is evidence that AKR1C2 expression is sex-dependent in several tissues, underlined by its involvement in progesterone and dihydrotestosterone metabolism [32,33,34]. Therefore, the impairment of sex-dependent turnover of these compounds might result in subsequent consequences for diseased tissue.

Although all other AKR1Cs are encoded on the forward strand, AKR1C2 is encoded in the opposite direction. However, this means that AKR1C2 shares gene regulatory elements with its neighbors, as shown for a cis-regulatory region common to AKR1C2 and AKR1C1, which raises the possibility of joint regulation [35]. 

The alternatively spliced version of the HPV16-E6 full-length protein (HPV16-E6*I) can directly promote AKR1C1 and AKR1C3 expression by binding to SP1 binding sites in their promoter regions [20,28]. Furthermore, E6*I promotes signaling pathways of the oxidative stress response, including the activation of NRF2 signaling [36,37]. Our observation that AKR1C2 is upregulated in a subgroup of HPV-positive OPSCC patients and is a strong indicator of prognosis suggests HPV-specific regulation. However, the overexpression of HPV16-E6*I did not alter AKR1C2 mRNA expression, as observed for AKR1C1 and AKR1C3, indicating an alternative HPV-induced regulation. Furthermore, AKR1C2 shows independent protein expression compared to its counterparts AKR1C1 and AKR1C3 [20]. AKR1C4 expression was not included, as it is reported to be exclusively expressed in a liver-specific manner and was negative in our previous analyses [20]. The AKR1C family enzymes including AKR1C2 are capable of detoxifying components of tobacco smoke such as nicotine-derived nitrosamine ketones (NKKs). Cells can protect themselves against external stressors by increasing enzyme expression. Additionally, it has been shown that AKR1C2 can also reduce chemotherapeutics such as cisplatin, which leads to cisplatin-resistant tumors [38]. However, a history of alcohol and/or tobacco consumption did not affect AKR1C2 expression levels or specific prognoses. 

AKR1C2 has been reported to not only affect overall survival by interacting with various metabolic pathways but also to act as an oncogene by activating the PI3K/AKT pathway [38,39], thereby inhibiting apoptosis and increasing proliferation. Furthermore, several HNSCCs carry cancer-associated mutations in the PIK3CA gene, which promotes signaling via the PI3K pathway and thus stimulates tumor cell growth [40,41]. PI3K signaling induces the production of cyclooxygenase-2 (COX-2) via immunosuppressive prostaglandin E2 (PGE2) [24]. AKR1C2, in turn, is involved in prostaglandin metabolism by favoring proinflammatory and proliferation-promoting prostaglandin F variants, thus inhibiting the apoptosis-promoting prostaglandin J2 [24].

Our observation that the subgroup of female patients presenting with increased AKR1C2 expression showed more favorable overall survival is consistent with recent findings of significantly better 5-year OS in women with HNSCC and in the OPSCC subgroup [27,42]. Different lifestyles, HPV status, immune responses, and hormonal influences were discussed as possible factors for these findings [43].

In regard to AKR1C1 and AKR1C3, it has been reported that they play a role in the metabolism of estrogen. This results in a feedback loop where estrogen increases NRF2 activity leading to increased AKR1C expression [25]. However, our findings indicate that AKR1C2 expression does not show a correlation with NRF2 expression, which suggests that it is not involved in this regulatory feedback loop.

However, E6-mediated repression of proliferator-activated receptor gamma co-activator 1α/estrogen-related receptor α (PGC-1α/ERRα) may contribute to the observed differences in AKR1C2 expression between sexes. Inactivating the PGC-1α/ERRα pathway results in a lower mitochondrial antioxidant capacity and, therefore, a reduced treatment resistance [26].

In recent years, interest in the pharmacological inhibition of AKR1Cs has increased because they catalyze key reactions in the metabolism of prostaglandins, steroidal hormones, and cytostatic substances, thus promoting the signaling pathways directly involved in oncogenesis. However, most of the available substances non-selectively inhibit all AKR1Cs. Important classes of such drugs are nonsteroidal anti-inflammatory drugs (NSAIDs), benzodiazepines, steroids, and flavonoids [44,45,46,47]. Interestingly, the most well-known NSAID, acetylsalicylic acid (aspirin), is known to exhibit potent inhibition of AKR1Cs [22]. Specific inhibitors of AKR1C1 (3-bromo-5-phenylsalicylic acid) and AKR1C3 (tolfenamic acid, indomethacin; phase I/II trial in prostate cancer, NCT02935205)) are also available. However, such a drug to inhibit AKR1C2 specifically is not available to date. However, the inhibition of AKR1C2 expression can be achieved by ursodeoxycholic acid (USDC), leading to a synergistic effect in cell lines when combined with cisplatin [38]. Nevertheless, some of these unspecific substances, including NSAIDs, are already therapeutically approved and therefore simply require a combination with well-known chemotherapeutic agents so they can be easily established clinically. The strategy of utilizing NSAIDs is supported by a study analyzing HNSCC with PIK3CA mutations or amplifications (which may implicate co-occurring AKR1C overexpression as already discussed) where regular NSAID use (≥6 months) markedly prolonged disease-specific survival [48,49] and the Nurses Health cohort study, in which the use of both aspirin- and non-aspirin-based NSAIDs prolonged the survival of ovarian cancer patients using the primary chemotherapeutic agent, cisplatin [50].

The present study is partly limited by the small number of cases included, particularly the number of female patients. Nevertheless, we obtained results comparable to those of previous studies with larger cohorts.

In conclusion, AKR1C2 expression in tumor tissue is sex-dependent and, therefore, has a different predictive value. Although increased expression in female patients is associated with a favorable prognosis, this is not the case for male patients. For this reason, future studies with (un)specific AKR1C inhibitors must consider the sex of patients.

## 5. Conclusions

This study provides an analysis of the expression and regulation of Aldo-Keto Reductase 1C2 (AKR1C2) in HPV-positive and HPV-negative oropharyngeal squamous cell carcinoma (OPSCC). The key findings indicate that increased AKR1C2 expression is significantly associated with positive HPV status and higher AJCC classification, reflecting its potential role in tumor progression. Notably, AKR1C2 expression exhibited a sex-specific prognostic impact, where high levels correlated with poorer outcomes in male patients but more favorable survival is suggested in female patients. This differential impact underscores the importance of considering sex-specific factors in OPSCC prognosis and treatment strategies. The implications of these findings are profound, particularly in the context of clinical and translational cancer research. The study highlights the need for personalized treatment approaches in OPSCC, potentially targeting AKR1C2, especially in HPV-positive cases. Given the association of AKR1C2 with oxidative stress mechanisms, the results also contribute to the broader understanding of how oxidative stress and detoxification pathways influence cancer development and progression. Despite these significant findings, the study has limitations that should be acknowledged. Further studies on larger cohorts with a higher proportion of female patients are needed to further substantiate the results presented here. Such studies may also demonstrate how an AKR1C2 evaluation can be integrated into routine pathological evaluation. Furthermore, additional research on AKR1C expression and inhibition by established pharmacological substances is warranted. In conclusion, this study emphasizes the critical role of AKR1C2 in the progression of OPSCC and its potential as a biomarker for tailoring personalized treatment strategies. The findings also highlight the complex interplay between viral infection, oxidative stress, and sex-specific factors in cancer biology, urging further exploration in these areas to enhance therapeutic efficacy and patient outcomes in head and neck cancers.

## Figures and Tables

**Figure 1 cancers-16-02976-f001:**
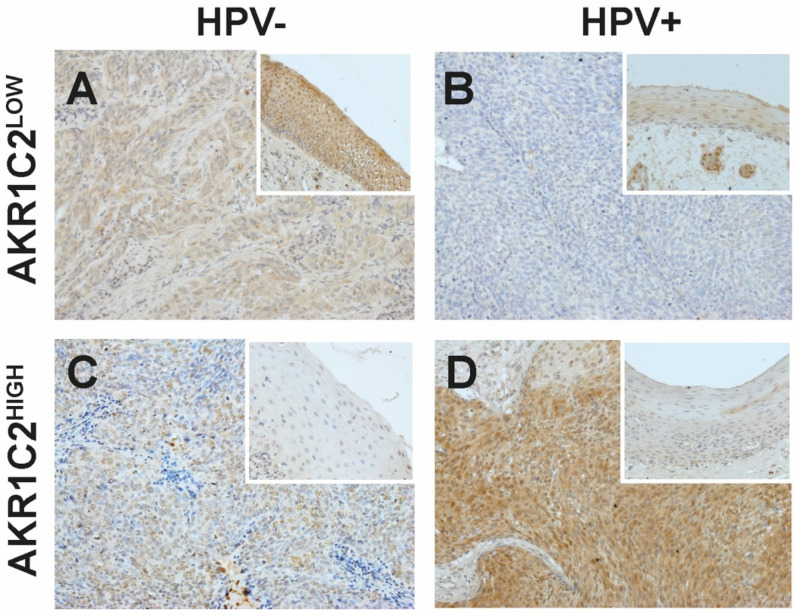
Representative immunohistochemical staining against AKR1C2 in non-tumorous (small rectangular images) and tumor tissue samples (**A**–**D**). AKR1C2^LOW^ indicates lower and AKR1C2^HIGH^ indicates higher expression of AKR1C2 compared to the adjacent non-tumorous epithelium. (**A**,**C**) HPV-negative (HPV−) OPSCC, (**B**,**D**) HPV-positive (HPV+) OPSCC. V = x200.

**Figure 2 cancers-16-02976-f002:**
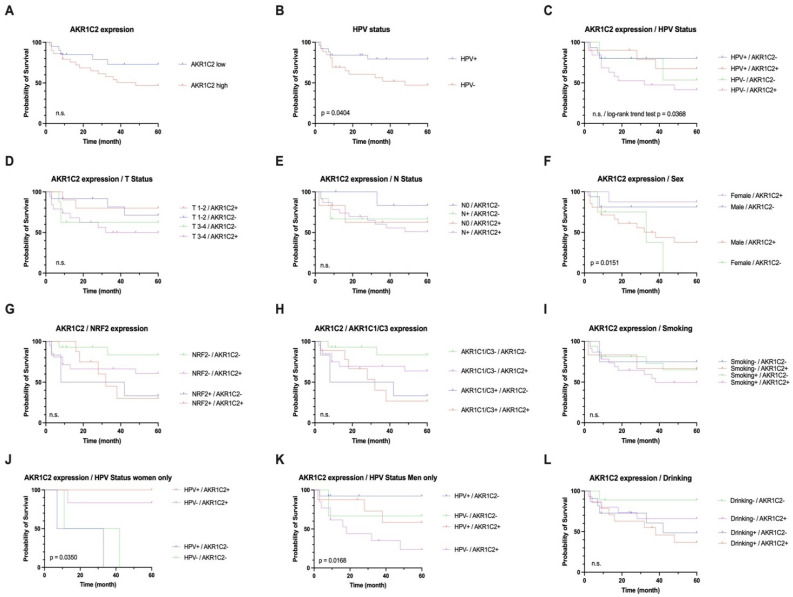
Univariate survival analysis for AKR1C2 expression status with low vs. high protein expression in tumor compared to adjacent non-tumorous epithelium. *p* value was derived by log-rank/Mantel–Cox test. Analyses of HPV status (**B**), AKR1C2 expression combined with sex, and combinations of AKR1C2 expression and HPV status in women (**J**) and men (**K**) proved to be significant.

**Figure 3 cancers-16-02976-f003:**
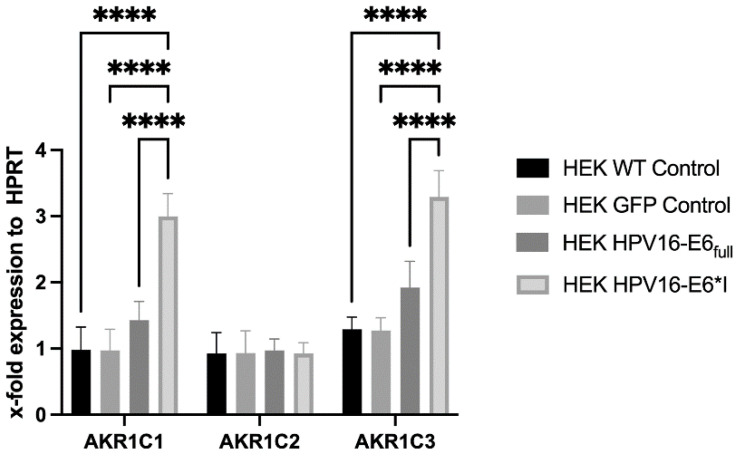
mRNA expression of AKR1C1, AKR1C2, and AKR1C3 in HEK293 cells overexpressing HPV16-E6 or HPV16-E6*I. Hypoxanthine Phosphoribosyltransferase (HPRT) was used for normalization of mRNA levels. (****) *p* < 0.00001.

**Table 1 cancers-16-02976-t001:** Summary of clinicopathological features of patients analyzed in this study.

				AKR1C2 Tumor Staining				
		Total ^(1)^		AKR1C2^HIGH (2)^		AKR1C2^LOW (2)^		
Clinicopathological Feature		*n*	%	*n*	%	*n*	%	χ^2^
**Mean age (years)**		51		55.125		60.162		
**Sex**								
	Male	39	76.5	21	42.9	16	32.7	
	Female	12	23.5	8	16.3	4	8.2	0.738
**T classification**								
	pT1 and pT2	23	45.1	10	20.4	12	24.5	
	pT3 and pT4	28	54.9	19	38.8	8	16.3	0.090
**N classification**								
	pN0	13	25.5	6	12.2	7	14.3	
	pN1–2 (3)	39	74.5	23	46.9	13	26.5	0.331
**M classification**								
	pM0	49	96.1	28	57.1	19	38.8	
	pM1	2	3.9	1	2.0	1	2.0	1.000
**AJCC classification**								
	I	14	27.5	5	10.2	8	16.3	
	II	12	23.5	4	8.2	8	16.3	
	III	10	19.6	8	16.3	2	4.1	
	IV	15	29.4	12	24.5	2	4.1	**0.009**
**Relapse**								
	Yes	23	45.1	13	26.5	9	18.4	
	No	28	54.9	16	32.7	11	22.4	1.000
**Death**								
	Yes	23	47.0	13	26.5	6	12.2	
	No	26	53.3	16	32.7	14	28.6	0.377
**HPV-status**								
	Negative	25	49.0	19	38.8	5	10.2	
	Positive	26	51.0	10	20.4	15	30.6	**0.008**
**Smoking**								
	Yes	40	78.4	22	45.8	16	33.3	
	No	11	21.6	6	12.5	4	8.3	1.000
**Alcohol**								
	Yes	25	49	14	28.6	11	22.4	
	No	26	47	15	30.6	9	18.4	0.773
**Localization**								
	Tonsil	31	60.8	16	32.7	15	30.6	
	Tongue base	15	29.4	8	16.3	5	10.2	
	Soft palate	5	9.8	5	10.2	0	0	0.122
**NRF2** **expression**								
	Nuclear	14	27.5	8	16.3	6	12.2	
	Cytoplasmic	37	72.5	21	42.9	14	28.6	1.000
**AKR1C1** **expression**								
	AKR1C1 (+)	15	29.4	20	40.8	6	12.2	
	AKR1C1 (−)	36	70.6	9	18.4	14	28.6	1.000
**AKR1C3** **expression**								
	AKR1C3 (+)	15	29.4	20	40.8	6	12.2	
	AKR1C3 (−)	36	70.6	9	18.4	14	28.6	1.000

*n* = Number of patients. Staging was performed according to AJCC/UICC 8th Edition in Oropharyngeal Squamous Cell Carcinoma. (1) Total number corresponds to the maximal number of patients analyzed. (2) Relative staining compared to normal epithelium. AKR1C2^HIGH^ means higher expression in tumor cells compared to normal epithelium, and AKR1C2^LOW^ means less or equal staining in tumor cells compared to normal epithelium. χ^2^: Chi-Square test for significance. For mean age, ANOVA is used to measure significance. Significant values are highlighted in bold.

## Data Availability

The data presented in this study are available upon request from the corresponding authors. The data are not publicly available due to ethical restrictions.

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
