# Peer review of "Analysis of Expression and Regulation of AKR1C2 in HPV-Positive and -Negative Oropharyngeal Squamous Cell Carcinoma"

_cancers, 2024, doi:10.3390/cancers16172976_

Round 1

Reviewer 1 Report

Comments and Suggestions for Authors

The manuscript by Ziogas et al. is an interesting paper about AKR1C2 regulation in HPV-negative and -positive oropharyngeal squamous cell carcinoma. The paper is well-written and interesting. In my opinion It is suitable for publication after some minor improvement:

line 53 the authors analyse epidemiological data that are too old. Please provide a more recent data and citation.

line 108-109 there is a typo

Reference are not in the style provided in the author's instructions

Author Response

We thank the reviewer for the time and effort to review our manuscript.

line 53 the authors analyse epidemiological data that are too old. Please provide a more recent data and citation.

We agree with the reviewer that the data cited is already quite old. In addition, the following sentence “The Global Cancer Observatory (GLOBOCAN) predicted an even greater increase in incidence, with an estimated rate of 30%.” also includes the aspect of increasing case numbers and is based on more recent data. We have therefore decided to delete the criticized sentence.

line 108-109 there is a typo

This typo was corrected.

Reference are not in the style provided in the author's instructions

References were updated according to the journal’s recommendations.

Reviewer 2 Report

Comments and Suggestions for Authors

Ziogas M et al., showed interesting study on the "Analysis of Expression and Regulation of AKR1C2 in HPV- positive and -negative Oropharyngeal Squamous Cell Carcinoma". I have some concerns as the following need to be addressed:

1- Female OPSCC samples need to be increased to fill up the bias of the comparison between the two sexes and provide strong evidence in AKR1C2 as the marker.

2- Add  the sequence alignment/similarity among ARK1Cs or cite on other published articles about the sequence alignment of AKRCs in the discussion section.

3- Detail about "*....*" in term of statistical significance in Figure legend 3.

4- Discuss more about why AKR1C2 is sex-dependence?

5- Did authors check the AKR1C4 expression in the investigated samples?

Author Response

We thank the reviewer for the time and effort to review our manuscript.

1- Female OPSCC samples need to be increased to fill up the bias of the comparison between the two sexes and provide strong evidence in AKR1C2 as the marker.

We agree with the reviewers 2 and 3 that the sample size of especially the female subgroup might be increased in further studies. However, the samples presented here corresponded to the patients treated at the University Hospital of Cologne during the specified period and were therefore not pre-selected. Therefore, as stated in the materials and methods section, 2.5 Statistics, we performed power analysis in advance and determined a minimum sample size for subgroup analysis of 8 samples giving a power of 90% and a significance level of 0.05.

2- Add the sequence alignment/similarity among ARK1Cs or cite on other published articles about the sequence alignment of AKRCs in the discussion section.

We added a citation to the relevant sentence in the Discussion section:

AKR1C2 is located in the same genomic region on chromosome 10p15-14 and con-tains all four Aldo–Keto Reductase Family 1 member C genes. The proteins show high sequence homology, namely AKR1C1/AKR1C2 with 98% homology, differing in only 7 amino acids, whereas AKR1C2/AKR1C3 show 87% homology and differ in 43 amino acids (Penning et al., PMID 30137266).

3- Detail about "*....*" in term of statistical significance in Figure legend 3.

We added this information to the figure legend.

4- Discuss more about why AKR1C2 is sex-dependence?

We added some more information to the Discussion:

Furthermore, there is evidence that AKR1C2 expression is sex depended in several tissues, underlined by its involvement in progesterone and dihydrotestosterone metabolism (PMID: 34439872, PMID: 15492289, PMID: 36768194). Therefore, impairment of sex dependent turnover of these compounds might result in subsequent consequences for diseased tissue.

5- Did authors check the AKR1C4 expression in the investigated samples?

We and others checked AKR1C4 expression in OPSCC tissue prior to this study and did not observe any hints for reasonable expression in this tissue, e.g. Huebbers et al., PMID: 30367463. See also proteinatlas.org.

Reviewer 3 Report

Comments and Suggestions for Authors

This study evaluates the expression of AKR1C2 by immunohistochemistry in FFPE samples derived from HPV-positive and –negative oropharyngeal squamous cell carcinomas.

The manuscript is very well written and the analyses described with the appropriate level of detail.

The study provides some interesting and clinically relevant findings, if they can be confirmed in larger, independent cohorts. An association between AKR1C2 expression levels and higher AJCC classification and HPV status seems to be reasonably well supported by the sample size and data. However, numbers are getting very small, especially in the subgroup of female patients (n=12), and for further subgroup analyses within female patients. For example, the authors state that “in women AKR1C2HIGH tumors showed beneficial survival regardless of HPV-status”; however, there were only 4 AKR1C2LOW tumors in this subgroup, 2 of them being HPV-positive and -negative, respectively (Fig. 2 J).

The authors may want to discuss in a little bit more detail what the small cohort size means for their conclusions, i.e. analyses in larger, independent cohorts may be warranted to confirm the initial findings reported in this manuscript etc.

Furthermore, the authors may also want to provide some more detail regarding potential observations made during interpretation of the AKR1C2 IHC staining patterns (given the complexity of the interpretation algorithm, which is based on relative staining intensities between cancer and adjacent normal tissue). Potential challenges associated with the proposed IHC scoring algorithm in routine clinical practice, especially with respect to potential IHC staining heterogeneity and cancer and/or normal tissue, and background staining, may be helpful to discuss.

Very minor: typo in line 109, “sthan”

Author Response

We thank the reviewer for the time and effort to review our manuscript.

The study provides some interesting and clinically relevant findings, if they can be confirmed in larger, independent cohorts. An association between AKR1C2 expression levels and higher AJCC classification and HPV status seems to be reasonably well supported by the sample size and data. However, numbers are getting very small, especially in the subgroup of female patients (n=12), and for further subgroup analyses within female patients. For example, the authors state that “in women AKR1C2HIGH tumors showed beneficial survival regardless of HPV-status”; however, there were only 4 AKR1C2LOW tumors in this subgroup, 2 of them being HPV-positive and -negative, respectively (Fig. 2 J).

We agree with the reviewers 2 and 3 that the sample size of especially the female subgroup might be increased in further studies. However, the samples presented here corresponded to the patients treated at the University Hospital of Cologne during the specified period and were therefore not pre-selected. Therefore, as stated in the materials and methods section, 2.5 Statistics, we performed power analysis in advance and determined a minimum sample site for subgroup analysis of 8 samples giving a power of 90% and a significance level of 0.05.

Nevertheless, we agree that the data presented in Fig. 2 J only show a trend highlighting the need for further studies with more female patients included. We therefore modified the mentioned sentence highlighting that only a tendency could be observed:

Whereas in women AKR1C2HIGH tumors presented with a tendency for beneficial survival regardless of HPV-status (HR = 0.333, 95% CI 0.028 – 3.977, p = 0.0350), HPV+/AKR1C2LOW tumors presented with far better survival probability, followed by intermediate outcomes for HPV+/AKR1C2HIGH and HPV-/AKR1C2LOW tumors (HR 2.300, 95% CI 0.734 – 7.201, p = 0.0168). HPV-/AKR1C2HIGH tumors presented with the most unfavorable outcome.

The authors may want to discuss in a little bit more detail what the small cohort size means for their conclusions, i.e. analyses in larger, independent cohorts may be warranted to confirm the initial findings reported in this manuscript etc.

We agree that future studies should highlight the aspect of AKR1C2 expression in female patients. We therefore added the following sentence to the Conclusion section:

Further studies on larger cohorts with a higher proportion of female patients are needed to further substantiate the results presented here.

Furthermore, the authors may also want to provide some more detail regarding potential observations made during interpretation of the AKR1C2 IHC staining patterns (given the complexity of the interpretation algorithm, which is based on relative staining intensities between cancer and adjacent normal tissue). Potential challenges associated with the proposed IHC scoring algorithm in routine clinical practice, especially with respect to potential IHC staining heterogeneity and cancer and/or normal tissue, and background staining, may be helpful to discuss.

We are already experienced in evaluating AKR1C1 and AKR1C3 expression as demonstrated in a series of publications (PMID: 30367463, PMID: 30367463). While the AKR1C2 expression pattern in general did not correlate with the other two proteins, as shown in this manuscript, protein expression was also pronounced in tumor and epithelial cells, but not in the surrounding connective tissue. When strong expression was observed, this presented in clearly definable cells such as lymphocytes or muscle cells (see results). Nevertheless, we agree that in routine pathology it might be challenging to evaluate this marker. Our group is experienced in AI analysis of pathological samples. We therefore plan to include AKR1C expression analysis because of their high relevance for prognosis and drug treatment in future projects. The following information was added to the Conclusion:

Further studies on larger cohorts with a higher proportion of female patients are needed to further substantiate the results presented here. Such studies may also demonstrate how an AKR1C2 evaluation can be integrated in routine pathological evaluation.

Very minor: typo in line 109, “sthan”

This typo was corrected. See also reviewer #1